# Experiences of adolescents and youth with HIV testing and linkage to care through the Red Carpet Program (RCP) in Kenya

**Judith Kose**[1,2☯‡]*, **Tyriesa Howard**[3☯‡], **Cosima Lenz**[4], **Rose Masaba**[5], **Job Akuno**[5], **Godfrey Woelk**[6], **Pieter Leendert Fraaij**[7], **Natella Rakhmanina**[4,8,9]

1 Africa Centres for Disease Control and Prevention (Africa CDC), Nairobi, Kenya, 2 Erasmus MC, Department of Viroscience, Erasmus University, Rotterdam, the Netherlands, 3 Brown School at Washington University in St. Louis, St. Louis, MO, United States of America, 4 Technical Strategy and Innovation, The Elizabeth Glaser Pediatric AIDS Foundation, Washington, DC, United States of America, 5 Country Program, The Elizabeth Glaser Pediatric AIDS Foundation, Nairobi, Kenya, 6 Research Department, The Elizabeth Glaser Pediatric AIDS Foundation, Washington, DC, United States of America, 7 Pediatric Infectious Diseases Division, Erasmus MC-Sophia/ Erasmus University, Rotterdam, the Netherlands, 8 The George Washington University, Washington, DC, United States of America, 9 Children's National Hospital, Washington, DC, United States of America

☯ These authors contributed equally to this work.
‡ JK and TH are first authors to this work.
* j.otieno@erasmusmc.nl

**Data Availability Statement:** To protect the privacy and confidentiality of participants in the study, the data will not be shared publicly. For this purposes, the contact for data requests is Stephen

## Abstract

Adolescents and youth living with HIV (AYLHIV) experience worse health outcomes compared to adults. We aimed to understand the experiences of AYLHIV in care in the youth-focused Red-Carpet program in Kenya to assess the quality of service provision and identify programmatic areas for optimization. We conducted focus group discussions among 39 AYLHIV (15–24 years) and structured analysis into four thematic areas. Within the HIV testing theme, participants cited fear of positive results, confidentiality and stigma concerns, and suggested engaging the community and youth in HIV testing opportunities. Within the HIV treatment adherence theme, participants cited forgetfulness, stigma, adverse side effects, lack of family support, and treatment illiteracy as barriers to adherence. Most participants reported positive experiences with healthcare providers and peer support. In terms of the HIV status disclosure theme, AYLHIV cited concerns about their future capacity to conceive children and start families and discussed challenges with understanding HIV health implications and sharing their status with friends and partners. Youth voices informing service implementation are essential in strengthening our capacity to optimize the support for AYLHIV within the community, at schools and healthcare facilities.

## Introduction

There are approximately 1.65 million adolescents (10–19 years) living with HIV globally [1]. Adolescents and youth ages 10–24 years are a uniquely vulnerable population and experience high HIV prevalence rates and suboptimal treatment outcomes [2, 3]. In Kenya, HIV remains

Siamba; Technical Adviser- Data management and analysis; Email ssiamba@pedaids.org; alternate email: Stephen.siamba@gmail.com; Tel: +254 714 864 693. In response to reasonable requests with the assigned contact person mentioned above, de-identified data may be shared in line with ethical approval. Stephen Siamba is the Data Manager/ Statistician on the study and as such is the custodian of the dataset. Additionally, The Kenya IRB - KNH-UON Ethics and Research Committee (https://erc.uonbi.ac.ke/) under whose jurisdiction the study was approved, is responsible for ethical restrictions on the (qualitative) data.

**Funding:** This study was supported with funding from a grant from the Positive Action for Adolescents Programme by ViiV Healthcare UK Ltd, London, UK under the Positive Action for Adolescents Program (https://viivhealthcare.com/ hiv-community-engagement/positive-action/). EGPAF's adolescent and youth-focused programs in Kenya are supported by funding from United States Government (USG), ELMA Philanthropies, and ViiV Healthcare UK Ltd, London, UK. The funder did not participate in the implementation of the program or interpretation of the results as presented in this manuscript. The results do not necessarily represent the views of the funder. The funders had no role in study design, data collection and analysis, decision to publish, or preparation of the manuscript.

**Competing interests:** The authors have declared that no competing interests exist.

a leading cause of mortality and morbidity among adolescents and youth living with HIV (AYLHIV) [4]. Poor linkage to and retention in HIV care, high loss to follow-up prior to anti-retroviral treatment (ART) initiation, and sub-optimal treatment adherence contribute to high adolescent HIV morbidity and mortality [5, 6].

The period immediately following the diagnosis of HIV is particularly important, and tailored interventions for newly diagnosed AY are essential to ensuring successful linkage to and retention in care. Several studies have identified promising practices to engage AYLHIV, including provision of youth-friendly services, individual and group counseling, peer support, and financial incentives [7, 8]. A series of qualitative evaluations in Kenya gathered important insights from AYLHIV and various stakeholders around adolescent-specific barriers impacting their linkage and retention in care. These barriers included stigma, poverty, fear of disclosure, mental health challenges, and limited social support, including non-supportive behavior by healthcare workers [9]. The challenges and complexities AYLHIV experience concerning disclosure of HIV status, stigma, and treatment management are well documented in the literature alongside the implications these experiences have on AYLHIV long-term engagement and retention in care [10]. In school settings, in particular, stigma perpetrated by teachers and peers has been described in published studies from Kenya, where boarding schools are common and many AYLHIV chose not to disclose their HIV status and treatment to the educators and school staff [11, 12].

Despite growing awareness of the barriers and promising research around facilitators, we are unaware of studies that have explored newly linked-to-care AYLHIV perspectives on their experiences while being linked to care and throughout the first few months following their diagnosis. AYLHIV perspectives are crucial and promote shared decision-making and capacity building, which generates evidence to improve healthcare and support services [13–15]. During the years 2016–2021, with support from ViiV Healthcare, the Elizabeth Glaser Pediatric AIDS Foundation (EGPAF) implemented the Red Carpet Program (RCP), designed to improve the long-term health outcomes of AYLHIV in Homa Bay County, Kenya. The RCP had originally conceptualized that providing support to newly diagnosed AYLHIV at the point of linkage to HIV care through reducing barriers and provision of tailored support, would reduce potential stigma and increase the likelihood of retention in care. This study sought to conduct a qualitative evaluation of newly linked AYLHIV experiences in care at RCP facilities across treatment domains to assess the quality of service provision and identify programmatic areas for optimization for this population [16].

## Methods

The RCP, launched in 2016, is a package of adolescent-specific services and interventions, beginning at the time of HIV diagnosis through the provision of fast-track, peer-supported VIP (very important person) services [16]. RCP facilitated comprehensive care at the facility and within the community, prioritizing bi-directional linkages with boarding schools to ensure uninterrupted care and support for AYLHIV in school [17]. We conducted a prospective qualitative study to evaluate the self-reported experiences among a convenience sample of AYLHIV who were in care at the RCP healthcare facilities through four focus group discussions (FGDs) with AYLHIV aged 15–24 years. We held two separate FGDs with adolescents 15–17 years of age and two separate FGDs with youth 18–24 years of age.

### Participants and procedures

The study participants included male and female AYLHIV (acquired vertically or horizontally) aged 15–24 years who were in care at one of the 60 RCP facilities in Homa Bay County, Kenya.

AYLHIV who attended at least one clinic visit in one of the study facilities and were engaged in care between 0–12 months prior to the study were eligible for FGD participation. Eligible AYLHIV were approached during routine clinic visits and invited for participation in FGDs by Red Carpet program coordinators and adherence counsellors using a standardized recruitment script. The target recruitment was 32–40 AYLHIV for four FGDs (8–10 per FGD, stratified by age groups 15–17 and 18–24 years).

Since participants were recruited from healthcare facilities located within 8 Homa Bay subcounties, we divided participants into 4 geographically convenient groups to conduct FGDs in proximity to their residence or healthcare service points outside of the facility. FGDs were held in the non-clinical spaces of the local healthcare facilities. The FGDs were located within confidential clinic-based spaces at four facilities in closest proximity to where participants lived. These included Marindi, Mbita, Ndhiwa, and Kendu Bay healthcare facilities. Indoor spaces with doors and comfortable seating were ensured to be private and secure prior to hosting the FGD. Additional information regarding the ethical, cultural, and scientific considerations specific to inclusivity in global research is included in the Supporting Information (S1 Fig).

During a weeklong workshop, the recruitment team and research assistants (RAs) were trained on the protocol and the FGD guide, (including translation for meaning from English to Dholuo), research ethics and on data capture. The training involved extensive time roleplaying FGDs and included probing techniques. The RAs and recruiters also signed confidentiality agreements.

Upon arrival for the FGDs, written informed consent was obtained for all enrolled participants. The demographic information and the timing of the HIV diagnosis were self-reported within an individual, anonymous questionnaire from each participant. The sex of the individual was asked in the questionnaire, and this data was reported at an aggregate level for each FGD. The designated EGPAF-trained FGD moderator or notetaker worked with illiterate participants to complete the survey verbally, as needed, and recorded the participant responses on the survey form. FGDs were conducted in the local Dholuo language, and were audio recorded on password protected tablets. Each FGD had a dedicated note-taker. Notes and audio-recordings were kept in secure locations in the field. Once the FGDs were completed, the FGD survey responses and the FGD notes were subsequently transcribed and used to generate summary field notes following each FGD. All notes and responses were translated (with accuracy checks) into English for data analysis. The principal investigator and the study coordinator reviewed each of the transcripts for completeness and accuracy. Identifiable data were not collected during the interviews; unique IDs were created for the participants and assigned at the time of the interviews. On return from the field, the encrypted and password-protected audio recordings were downloaded to and stored electronically on the study server. The notes and informed consent forms were stored separately in lockable cabinets with restricted access.

Transcripts were translated in an anonymous format by certified interpreters. RAs reviewed the transcriptions of the FGD recordings and translated them into Microsoft Word. Two RAs, under the supervision of a lead qualitative scientist, generated a code list using a combined inductive and deductive approach. Codes were identified a priori from the FGD guide while reviewing data allowed unanticipated codes to emerge. Transcripts were uploaded into the MAXQDA and coded by two RAs. The code list was updated several times, and code definitions were further refined as emergent codes arose during the coding process. Data were summarized through descriptive, text-based summaries and data display matrices. Data were analyzed by study population groups (15–17 and 18–24 age groups), comparing similarities and differences across the groups. Related codes were grouped into overarching themes. Data reduction and summary tables were created to organize the results and track the emergence of overall themes and key findings. Quotes were selected from the data to illustrate thematic findings.

Research materials can be found in the Supplemental Information (S1 Appendix).

FGD gathered participants' opinions and perceptions in the following thematic areas: 1) HIV testing and linkage to care; 2) retention in care and adherence to treatment; 3) HIV support services (including peer counseling and peer groups); and 4) HIV status disclosure.

### Ethical approval

This protocol was approved by the Kenyatta National Hospital Ethics Research Committee, Kenya, and the Chesapeake Institutional Review Board, USA. Written research consent was obtained from each participant ≥18 years of age prior to conducting FGDs. For participants aged 15–17 years, parental/guardian consent and AYLHIV assent were obtained. A waiver of parental or guardian consent was granted for mature minors 15–17 years old (married, pregnant, having children, or being a head of a household) [14].

## Results

A total of 39 AYLHIV were recruited and participated in four FGDs (18 participants in two FGDs for 15–17 years old and 21 in two FGDs for 18–24 years old). The FGDs included 26 females and 13 males. The majority (72%) were female; the median age was 18.3 years. The majority (75%) of the respondents reported being currently in school. The mean self-reported length of knowledge of HIV status was 8.7 years; all participants reported being on treatment. Below, we summarize the FGD results by thematic areas.

### HIV testing and linkage to care

FGD participants described that most adolescents have a general understanding of the importance of HIV testing. Most participants described the fear of receiving a positive HIV test result as a contributing factor in the general reluctance to seek HIV testing among adolescents. Other significant barriers to HIV testing identified were HIV-associated stigma and youth not wanting to be tested at the same facilities as their peers. In addition, certain testing practices were described as problematic, especially among adolescent males. One participant described how "boys test less frequently than girls" out of fear or reluctance to visit HIV testing facilities. This participant (15–17 year old, Marindi) described that when testing is made readily available at tents within the community during "market days," boys also often avoid these public opportunities for HIV testing, indicating a preference for more discreet testing opportunities. In general, participants considered adolescent females to be more focused on the physicality of their bodies and concerned about pregnancy-related health considerations in comparison to adolescent males as a plausible contributor to their higher HIV testing rates (Table 1). One participant discussed how girls may discover rashes or other disease manifestations on their bodies and subsequently seek medical advice. Participants further described adolescent males as more nonchalant or less likely to give considerable care to any disease manifestation. Or, if adolescent males acknowledged health issues, they were described to more frequently default to self-medication alternatives, including alcohol use, rather than seeking medical care. FGD participants also discussed that while testing was sometimes offered at schools, significant concerns about privacy and confidentiality and the risks of teachers and school staff potentially disclosing their status exist among youth and frequently prevent them from accessing HIV testing.

The intricacies of interpersonal relationships were also described as barriers to testing among sero-discordant and unaware of their HIV status partners, especially among HIV-positive adolescent females and their HIV-negative adolescent males' partners who were reluctant to get tested. One participant (15–27 years, Mbita) described his experience as following:

**Table 1. AYLHIV quotes on identified themes.**

| HIV Testing and Linkage to Care |
|---|

**Motivation to get tested**

"I went to hospital. I decided to go and get tested that day and when I reached, I found the nurse and I told her, 'Nurse, my kids have already died and told there is someone who has told me HIV/AIDS is the disease killing my children.' So, I have gotten pregnant. I have decided to come and get tested."– 18–24 years, Ndhiwa

"What I would encourage my fellow youth to go for testing is that, you go for testing because you are concerned about your life. That's the first one. The second one you go for testing because you are concerned about other people's lives that are near you. Suppose you are married, you are concerned about the life of your partner that you have."– 18–24 years, Ndhiwa

**Reluctancy by male partner to be tested**

"When I realized my status, I had a boyfriend, and when I told him that I was positive, he doubted and told me that I was lying. But he really fears hospitals. I told him, 'I have been found positive,—do you think you are also sick? But he told me he was not sick. He just feels okay and he never falls sick. In case he is sick, he visits a chemist [local herbalist] to buy drugs [medication]. He has never visited a hospital. Even those who walk from door to door have never treated him. There was a day I went to the facility and one nurse asked me about him, and I was free and told her how he responded. She sent me to go and talk to him. He was called but up to date he has never gone. . . ."– 15–17 years, Mbita

**Awareness among the youth**

"Most teenagers are positive, but they are not aware. This is because they would prefer to test when they fall sick. In fact, my cousin will not even stay behind when they [healthcare professionals] pay a visit [within the community]. She won't test [for HIV]. She would say she is going to test only when she gets married. And in fact, when we were in the 'Dream Girls' [a community project] she was never tested. Yet, her parents are all positive. Maybe she thinks if her parents are all positive, maybe she is also positive. So, it's all about stigma that is hindering them. They need some counseling on the importance of starting medication at an early stage before the multiplication of the virus."– 15–17 years, Mbita

**The importance of caregiver and community education about testing**

"At the time of testing I was living with a guardian. The guardian [saw] that I was constantly sick with wounds all over the body and not healing after treatment. They got tired and decided to take me for testing. My parents had died, but nobody knew the ailment that killed them, so my testing is what made them realize that my parents died from complications of HIV/AIDS."– 18–24 years, Kendy Bay

**Male engagement**

"My opinion is that boy children should be ambassadors of change among the boys. For example, when we conduct outreach activities, the boys should serve as an example. When they go for outreach, the boys will see that one of their own has been tested and knows his status. This will encourage them to get tested. They should be ambassadors of change among the boys." -18-24 years, Kendu Bay

**Testing during pregnancy**

"For me, what I wanted to ask, because I was tested when I was pregnant, was, 'Now that I am pregnant and I have already been tested and found with HIV, [will] my unborn baby be found with the disease?"– 18–24 years, Ndihwa

| Retention in Care and Adherence to Treatment |
|---|

**Delayed start of treatment**

"When I was tested, I was still young and did not understand much, but could grasp a few things. When I was taken to the first facility, my HIV test results came back positive. The providers refused to enroll me in treatment because I was too young. Then, I was taken to another facility, and the providers tested me and [again] refused to enroll me, saying the same thing, that I was too young—I cannot use the medication. At that time, both my parents had died. I was living with a guardian, so when I came to another place, they tried talking to me, given that my guardian was pleading with them (while) crying, (that) they had to enroll me on treatment."– 18–24 years, Kendu Bay

**Stigma of ART**

"Going to pick your drugs, you get other patients coming for other ailments, so you will feel stigmatized. Just seeing that 'now this person has seen me taking these ARVs'. . . so this is encouraging stigma."– 15–17 years, Mbita

"If there are visitors in the house and your mother calls you out loudly to go and take your drugs. . .yet your other siblings are not. This can make you feel bad."– 15–17 years, Mbita

**Confidentiality concerns**

"If you're going for the drugs [medication], the person behind you will always know you're going to take them. When your pills are counted, he/she just stands there and watches." - 15–17 years, Mbita

"These bottles where we store drugs in; their design should change. Like right now, everybody knows that this is the bottle used by those who are sick. They are put in a certain design whereby everybody who comes to the facility goes home with."– 15–17 years, Marindi

"When a friend suddenly comes to visit and the time to take drugs reaches, you have to wait until the friend leaves, and this will mean your time [to take medication] has passed."– 18–24 years, Kendu Bay

**Religious Barriers**

"Some religion[s] believe that drugs [medications], such as those for HIV, were brought with the devil to drink blood from people. Just myths and misconceptions so if you are in that religion and a staunch follower of their doctrines you stop taking drugs [medication] immediately."– 18–24 years, Kendu Bay

**Self-motivation**

"What motivates me is that when I commit to taking drugs I do not look any different from those who are not taking drugs, those who are HIV negative. We are just the same, there is no difference."

"What motivates me to take my medication first is to take medication is my future ambitions and the need to achieve them. Second, is in order not to look like those people who are HIV positive but refuse to take their medication."– 18–24 years, Kendu Bay

"The dreams we have as youths, the need to achieve them is what motivates me to take medication, and it is said when you have been tested and found HIV positive it is not the end of life. That being the case, it is motivational enough for one to keep taking drugs every day so that you do not die before your days. As long as you adhere to drugs, HIV will not kill you."– 18–24 years, Kendu Bay

"What motivates us is when you go to the hospital and you're tested and they find your viral load to be down, you will continue taking them so that you maintain them at that level."– 15–17 years, Mbita

(*Continued*)

**Table 1.** (Continued)

| HIV Testing and Linkage to Care |
| --- |

**Future treatment options**

"Some people go missing drugs, but if there is injection after every month this can help those who don't adhere to drugs."– 15–17 years, Marindi

"The reason I see injection as better, it will take a long duration before you go back for another injection. That is better than taking drugs every day. Sometimes one has so many things to do in the house that they even forget to take drugs [or] just remember later when the time has passed. With the injection you just know off head your next injection date. It is better than taking drugs."– 18–24 years, Kendu Bay

"Injection is better but when you want to stop having your ARVs and starting the injections, how can they commence it? How can it be done so that you leave ARVs and move to injections?"– 15–17 years, Mbita

| HIV Support Services |
| --- |

**Initial counselling**

"When I was tested and turned positive for HIV it was not easy for me. I was counseled well. I was very worried how I was going to start the medication, how I was going to tell my mother that I was HIV positive, I was worried of her reaction." - 18–24 years, Kendu Bay

"After receiving my diagnosis, I just felt alright because of the counseling I received, this is what made me start on HIV medication immediately."– 18–24 years, Kendu Bay

"The nurse that tested me, she talked to me in a friendly manner and with the words from her mouth and the way she was behaving, I could not miss taking drugs."– 18–24 years, Ndhiwa

"Ok, according to my views, they were harsh, did not mind what they were telling you, for example saying "how will this child be given the medication, if they have HIV they should go away and die" so I was left for dead with no future."– 18–24 years, Kendu Bay

**Supportive healthcare workers**

"If I go to the sister who gives us drugs, she first sit(s) down with you and ask(s) how you are using drugs [medication]. If you are genuinely using drugs [medication]. Then she again encourages you through training on how you should take drugs [medication. You don't stop taking drugs [medication]. Ever since I started coming to the clinic, I feel that I have a healthy life. I feel good because when I first went to the hospital, my viral load was high, but now low."– 15–17 years, Marindi

"When I started taking these drugs, I was advised [to] eat something after swallowing it to reduce the impact of these drugs. Because sometimes when you go to school and you didn't eat, you mostly feel tired, you cannot even play with your colleagues."– 15–17 years, Mbita

"What the sister told me is that when you have been found to be HIV positive and started taking drugs you are like someone who has stepped on the head of a snake and holding it down, the tail will keep struggling for it to be released in order that it can harm you but because you are stepping on the head it cannot bite you and harm you but when you stop taking your medication, you will have released your foot from the snake's head and it will bite you and harm you, the story meant that I should not stop taking my medication, if I stop I will either be harmed by diseases or even die."– 18–24 years, Kendu Bay

**Importance of Peer support**

"The people that are supposed to be counselling and training the adolescents should be an adolescent, not an adult, who the adolescent cannot relate to or open up to about the problems they have."– 18–24 years, Kendu Bay

"Adolescents need their time and date and privacy when going to take medication. They should also have a peer educator of their age because there are things a youth cannot open up about to an adult. It is only another youth who can understand the challenges they go through. They can have a discussion and reach a solution."– 18–24 years, Kendu Bay

"I like the part where peers come to teach us how we can take our medication and adhere to them and maintaining the clinic date visits."– 18–24 years, Kendu Bay

"They [peers] counselled me well, and they told me that if I accepted to take the medication, then I will be able to prolong my health but if I don't, the virus is going to harm my life. So I personally decided to go and take those drugs in order to prolong my life."– 15–17 years, Mbita

"After they had tested me and gave me the results, the peers took me to a private room where I received counselling, was told the importance of taking medication and adhering."– 18–24 years, Kendu Bay

"What I like is the time when we sit down receive counselling and share our issues as adolescents. That is the time I am able to share my problems."– 18–24 years, Kendu Bay

"The HIV positive peers can talk to them, telling them the importance of staying in drugs and that having HIV is not the end of everything just giving them courage."– 18–24 years, Kendu Bay

"Young people are motivated to stay on drugs [treatment] to the extent that their viral load is suppressed, but before there were no peers who motivated the young or visit them."– 18–24 years, Kendu Bay

**Flexible youth-friendly confidential services**

"The time I started taking my drugs, they used to mix both the adolescents and the adults. You could imagine if the following day was your clinic day and going into the same facility with certain adults who could spot you. This was making me go very early or late in the evening. When EGPAF called us to Homa-bay Twin Towers hotel, I mentioned it. When I came back for my clinic the next time, I found that we, the adolescents, will be coming on a Thursday and the adults on Monday or other days."– 15–17 years, Mbita

"There was also another problem in the pharmacy. You'd wait at the pharmacy for long yet he [the pharmacist] is not there. This also we raised [as a concern] and it was resolved. In fact, if you go there, at the pharmacy today, you will find him/her waiting for you even if there is nobody there. You get in, get served, and leave. Again, we were moved from the public and taken up somewhere hidden room. Nobody realizes what has brought you there. You finish faster and go back."– 15–17 years, Mbita

| HIV Status Disclosure |
| --- |

**Disclosure of perinatal HIV infection**

"I asked myself: Where did I get HIV/AIDS? I went and asked my aunt because she is the one that act as my mother. She told me that I got HIV/AIDs when my mother was breastfeeding me. That's where my aunt tried to encourage me."- 15–17 years, Marindi

(*Continued*)

**Table 1.** (Continued)

| HIV Testing and Linkage to Care |
| --- |

**Disclosure at school**

"She [the teacher] took those drugs. One day she took my drugs to the principle. Then, I was called and I disclosed to all of them. I came with my brother and disclosed to her and the principle. Yes, I was forced to disclose."– 15–17 years, Marindi

**Disclosure at home**

"I felt bad in my heart that all my siblings don't have it. Yet, I have this disease. It's imagining that it is only me who will be taking these drugs in the family. This was difficult for me. Initially, I was not used to this but later I got used to it."– 15–17 years, Mbita

**Delayed disclosure**

"I started [treatment] a long time ago and I used to find it so difficult because I did not know the reason why I was taking drugs because those days I was still too young. So, I used to ask myself why am I taking drugs? and I was told its TB. So, you know when I am in school, we are taught things like Tuberculosis and other diseases and for this one we are told this one for 6 months it will end. So, you know, I counted 6 months. When it was finished, I went back and told them 6 months is over, if it's TB why are you still giving me drugs."– 18–24 years, Ndiwa

**Disclosure in the community**

"So, I do not know how this thing spread. After it was known, I told my mother that no need to continue taking drugs here. I felt like changing the place where I get my drugs. Because this makes me collide with so many people."– 15–17 years, Marindi

**Disclosure by medical provider**

"I went to the hospital and asked the doctor, I am just realizing that I am taking some of the medicines and I don't know why I am taking them. So, he talked to me, counselled me about HIIV/AIDs. Ways I can get HIV/AIDS."– 15–17 years, Marindi

**Disclosure to multiple people**

"I have disclosed my status to so many people. For example, as girls, when a boy comes to seduce [court] us, the first thing is disclosure because you can be in a relationship with a boy without disclosing your status. You will be risking their lives, because he might be negative and you are positive and you might have sex without using condoms. If you are not adhering to drugs there are high chances you will infect the boy, so I disclose to boys who seduce [court] me."– 18–24 years, Kendu Bay

"When I realized my status, I had a boyfriend; and when I told him that I was positive, he doubted and told me that I was lying. . ."(Table 1). In this case, despite the adolescent woman's awareness of her HIV status and disclosure and encouragement for her male partner to get tested for HIV, it never took place. Fear and denial of knowledge of HIV status were cited as barriers to getting tested by AYLHIV who had HIV-positive parents. One participant described an adolescent female with HIV-positive parents who refused to learn about her HIV status (Table 1).

In response to the barriers to HIV testing that were identified, the participants described a series of strategies that could potentially help with engaging adolescents in testing. Educating the community about the indications and the importance of testing for HIV was identified as an effective strategy, supported by the experience shared by one of the participants on their experience with the diagnosis of perinatally acquired HIV (Table 1).

Engaging youth in HIV testing was also described as essential to successful HIV testing. Participants highlighted the importance of establishing a general awareness of the importance of HIV testing among youth. One participant (18–24 years, Kendu Bay) suggested: "We can create outreach. If the youth know that there is free treatment given during outreach, they will come, and the outreach should have HIV testing services." Participants also proposed promoting HIV testing among youth involved in romantic relationships. They suggested youth should engage in sexual health discussions within relationships and encourage partners to seek HIV testing. In addition, participants also suggested these sexual health dialogues could be enhanced by educational forums that described the benefits of HIV testing. Another strategy to better engage adolescent males in HIV testing centered on their interests in sports and suggested introducing a sport event or tournament whereby HIV testing was readily available. A few participants also suggested engaging adolescent males in testing by coupling it with other testing such as malaria testing, during an outbreak. Some participants felt the benefits of HIV testing should be sufficient self-motivation for adolescents to engage in testing (Table 1).

Participants also discussed how peer advocates could potentially contribute to favorable testing practices. Some suggested that using themselves as examples could increase uptake in testing behaviors. One participant (18–24 years, Kendu Bay) shared, "Being an HIV positive youth, you open up to them, telling them that you are taking HIV medication. They can listen to you and agree to get tested. During door-to-door testing, they normally hide; so, you can be open with them and convince them to get tested." In addition to general peer support for HIV testing among all adolescents, one participant felt that peers would be particularly helpful for engaging adolescent males in testing (Table 1).

## Retention in care and treatment adherence

Most FGD participants cited HIV treatment adherence barriers, such as general forgetfulness and dealing with internalized stigma associated with being the only HIV-positive person in their family. Some participants provided examples of how arriving home later than anticipated could contribute to them missing a designated dose, which would result in their decision to skip the dose altogether and then take multiple doses at once. Many participants agreed that side effects of HIV and antiretroviral treatment, including sickness, skin rashes, or skin discolorations, were potential barriers to treatment adherence. In addition, traveling and not having medication or being in an environment where people were unaware of their status also limited their ability to successfully take medication as prescribed. Furthermore, not having a reminder device, such as a phone or radio with an alarm, was also reported as a barrier to routinely adhering to treatment.

Participants cited a lack of disclosure of HIV status and not understanding the reason for treatment as an adherence barrier (Table 1). Furthermore, lack of family support, and living in shared spaces at boarding schools, and consequent fears of disclosure of HIV status among peers were cited as reasons for non-adherence. For example, participants cited the noises from pills rattling in bottles, drawing attention, and questions from peers.

In addition, selected participants noticed that having busy or conflicting schedules or competing priorities impacted their ability to attend clinical appointments. For example, asking for passes from teachers to leave class was particularly challenging for youth that had not disclosed their HIV status at school. For some, poverty was described as a barrier and not having sufficient food to eat in an effort to stifle medication side effects was related to not taking HIV medications. A few participants also described cultural practices and religious beliefs that they observed as barriers to treatment adherence (Table 1).

Participants described being educated about the Operation Triple Zero approach—zero missed appointments, zero missed treatments, and an eventual zero viral load—as the primary motivator for treatment adherence. They described striving to adhere to strict daily regimens of taking medications exactly within 24 hours and noted that strategies to reduce side effects were important (Table 1).

When considering the future HIV treatment options, many AYLHIV expressed interest in long-acting injectable HIV treatment (Table 1). Long-acting injections were also considered more confidential and less conspicuous than medication bottles as described by one participant (18–24 years, Kendu Bay): "It would be good because you are going to be injected at the facility. This will reduce the fear of taking drugs [and] the challenges we had talked about. [Such as] your friends visiting and you are afraid to take the drugs as when time reaches. The injections will eliminate such scenarios because when one goes to the hospital to get an injection, nobody will know the kind of injection you have received."

At the same time, some participants questioned the potential ramifications of transitioning from pills to injections (Table 1). Potential implications of transitioning to injections shared

by participants included the perception that individuals would be less inclined to return to the facility for supplemental treatment injections due to a misguided belief that they were now fully protected. One participant (15–17 years, Mbita) shared, "I think injection is the best, but it might bring in some laziness. You see even those going for depo, the family planning sometimes fails to go back because they now see themselves safe. It brings laziness because you're now free. But for ARVs you can see, the container is now empty let me go and pick more drugs." Other participants also discussed potential limitations with injections concerning the general fear sometimes associated with needles and described potential issues with participants forgetting to return to clinics for follow-up injection visits.

## HIV support services

Most FGD participants felt their care experiences were generally good and that healthcare providers educated them about the treatment; however, some participants reported feeling stigmatized by healthcare providers and also expressed concerns about confidentiality while receiving their care at the healthcare facility (Table 1). They described the RCP facility nurses as polite and felt the doctors were very engaged when speaking with youth (Table 1). Storytelling and providing relatable, vivid examples were described as effective strategies used by healthcare workers to promote treatment adherence (Table 1). Some participants also noted that employing more nurses could potentially contribute to quicker service delivery. Furthermore, continuing to simplify and speed up the dispersion of ART medications was also suggested. This was further extended by discussing efforts that could potentially reduce stigma around HIV services, which included separating HIV clients from other outpatient services, changing meeting days for the adolescents to avoid the adults, and ensuring that pharmacists are readily available. Emphasis was placed on the significance of having private entrances for AYLHIV and specific days for youth services at RCP healthcare facilities (Table 1).

FGD participants positively commented on their experiences with the support of the RCP community health volunteers. Participants described RCP representatives as *"good people"*. Participants shared that the discussions about treatment adherence during peer support groups were especially helpful for addressing barriers to adherence (Table 1). Participants also discussed the importance of overall education on sexual health, including safer sex practices, to support retention in care. Opportunities for advanced preventative health measures, including pre-exposure prophylaxis (PrEP), for partners were also discussed within the context of relationships. One participant (18–24 years, Kendu Bay) stated: "When you are living a positive life with HIV when we go to the clinics we are told that if we have partners and sometimes they are HIV negative, we [should] use condoms during sex. There are new drugs also called PrEP [and] the person who is HIV negative can use the drugs to prevent infection especially if married."

To improve adherence AYLHIV suggested expanding peer activities to trips and offering financial support to be provided as incentives to virally suppressed youth, as described by one participant (18–24 years, Kendu Bay): "I think they should introduce trips for those who are virally suppressed. This will encourage others to adhere to drugs in order to suppress the virus. Financial support for those still going to school, would be great, if that is possible." AYLHIV also felt that it would be increasingly helpful if other youth were more actively involved with counseling support services.

## HIV status disclosure

Several participants, who have self-identified as living with perinatally acquired HIV, discussed their experiences with parents or caregivers disclosing their HIV status to them. Those

participants shared their feelings and thoughts at the time of their HIV status disclosure and cited lack of understanding of HIV, since they were not able to visually see HIV symptoms and expressed concerns about their ability to have children and their family. Participants shared that finding out about their status was occasionally connected with learning about a relative's HIV status (i.e., a parent also living with HIV). Participants shared how disclosing caregivers helped explain how adolescents acquired HIV and talked about their future lives (Table 1). Participants also described their struggle with the notion of being the only HIV-positive person in their family (Table 1).

Following the disclosure of their HIV-positive status, participants described their experiences sharing their status most often with a close friend. However, they did not share their status with people whom they thought would spread gossip concerning their health, including certain family members as well. In addition to concerns about disclosing to family members, participants felt it was particularly important to disclose to romantic partners.

Participants also described how certain school policies impacted their HIV disclosure and access to medication. Many schools had "drug-free policies," which also included prescription medications. These types of policies often required students to present their medication to school personnel, which forced the disclosure of their status (Table 1). Considering school disclosure, participants described how schoolmates sometimes became suspicious of their status based on the attention they received from school nurses when asked to visit administrative offices to take treatment or if they were frequently excused from classes for clinic visits.

## Discussion

Our study provides valuable feedback from AYLHIV who received care through RCP in Kenya on the barriers and facilitators they face in testing for HIV, staying connected with HIV services, adhering to treatment and care, and undergoing disclosure of HIV status [16, 17].

External and internal HIV stigma were identified as challenges across multiple domains: hindering seeking HIV testing, particularly for adolescent boys and preventing treatment adherence, particularly in boarding schools. Similarly, studies of factors impacting HIV treatment adherence among AYLHIV in sub-Saharan Africa identified stigma as a main barrier to care and treatment adherence in home and school settings [18–21]. As described by our FGD participants, compared to girls, adolescent boys, are generally more uncomfortable seeking medical services until feeling quite ill [22, 23]. HIV and health services should take these gender differences into account when designing and implementing HIV testing and care services for adolescents and youth.

Like other studies, multiple adherence barriers were reported by ALHIV in our FDGs including forgetfulness, difficulty maintaining a treatment routine with changing schedules, experiencing side effects from the medication, and confidentiality [18, 21, 24]. Lack of family support was also cited as a reason for non-adherence among FDG participants. Social support from caregivers and/or family has been shown to be associated with self-reported ART adherence in studies from Uganda, South Africa, and other sub-Saharan countries, highlighting the importance of educating and involving caregivers and relevant family members in supporting AYLHIV in their treatment and care [18, 25, 26].

Disclosure of HIV status among perinatally infected AYLHIV remains a significant challenge worldwide, highlighted by many AYLHIV in our study. The proportion of disclosed children and adolescents living with perinatal HIV younger than 19 years of age in resource-limited settings ranges from 0 to 69% [27]. In Kenya, perinatal HIV disclosure rates have been documented as low as 36.6% among AYLHIV [28]. Data from Kenya and other sub-Saharan African countries suggest that disclosure of HIV status to perinatally-infected AYLHIV is

associated with improved treatment adherence and immunological status [10, 29, 30]. Similarly, in our study, disclosure was described to help AYLHIV understand the diagnosis and need for treatment. Similar to other studies, fear of stigma and rejection following disclosure with peers and in school settings were highlighted as significant challenges [31–33]. Concerns of rejection associated with stigma have been described by young people living with HIV as a factor which contributed to not disclosing their status [34]. Moreover, school-based challenges, such as stigmatizing "drug-free policies" described by our FGD participants, can be harmful in forcing disclosure of HIV status and deterring treatment adherence and retention in care. Such policies, while well intended to protect pupils, limit the AYLHIV choices for safe space to keep medications in dormitories, do not allow them to leave classes for care needs, and do not allow them to seek private space to take medications [35].

One participant in an FGD also observed that some boarding school environments limit capacity to adhere to treatment, due to the shared living spaces and difficulties hiding pills. This raises concerns about confidentiality and stigma among peers and teachers. Similar challenges have been reported by qualitative studies exploring barriers to ART among AYLHIV in boarding schools in other sub-Saharan African countries, such as Uganda and Kenya [11, 36, 37]. Our data and these studies highlight the unmet need for non-stigmatizing school environments and staff to support AYLHIV. Based on the feedback from our FGDs, we developed a set of RCP tools for the HIV care-friendly school environment and implemented them within the boarding schools in Homa Bay and Turkana counties in Kenya, which resulted in improved rates of AYLHIV engagement and retention in care [16, 38].

The concerns for confidentiality and stigma are not limited to school environments and extend into discussions around receiving care at healthcare facilities in our FGDs. Supportive interactions with healthcare providers in a respectful and informative manner have been shown to be associated with positive experiences with care among AYLHIV in Kenya and facilitate AYLHIV desire to return for services [39]. Concerns for confidentiality and fears of accidental disclosure to peers or community members by being seen at the facility are common among AYLHIV [40]. Within RCP healthcare facilities, we provided the AYLHIV with a responsive environment by implementing healthcare facility packages of services centered on AYLHIV needs and concerns, such as the ability to enter a private youth-friendly space avoiding wider clinic appearances [38]. Our FGD participants reported feeling welcomed at the facilities and described them as adolescent-centric and friendly. Similarly, other studies reported youth friendliness of services to influence ALYHIV uptake, satisfaction, and retention in care [41–43].

The importance of systematic peer support, and the availability of peer support groups at the healthcare facilities have been highlighted by AYLHIV in our study. ALHIV described the importance of being able to connect with peers and motivate one another. Participants emphasized the benefit of more intentional inclusion of youth in counselling and support services. Interaction with peers has been reported by AYLHIV to be enjoyable and encourage regular attendance at the clinic in other studies [44]. Story-based activities, which are often included in peer support groups, described by our FGD participants as adherence motivators, have been associated with a positive influence on the AYLHIV feeling of being isolated in sharing common experiences [45]. The benefits of peer support in improving adherence, retention and viral load suppression, and decreasing stigma have been reported in several qualitative assessments among AYLHIV from Uganda, South Africa and Kenya [9, 46, 47]. Data from our RCP peer-designed and supported model of care have equally shown increased rates of retention in care among newly diagnosed AYLHIV following new diagnosis of HIV [16].

AYLHIV in our study also provided insightful suggestions on improving the cascade of care, including streamlining, and optimizing the delivery of services while optimizing stigma-

reducing efforts. Integrating HIV testing into a multi-disease approach, as suggested by our AYLHIV participants, has been shown to increase efficiency while reducing stigma in sub-Saharan Africa (SSA), especially among men [48, 49]. When discussing future options of HIV treatment with long-acting antiretroviral drugs, AYLHIV expressed interest in the injections, citing potential benefits including saving time, improving adherence, and enhanced confidentiality. At the same time, concerns around fears of needles, forgetting to come back for injections, and confusions around the transition to long-acting antiretroviral drugs were voiced. In a US-based evaluation, YLHIV expressed high willingness and enthusiasm for long-acting HIV treatment, echoed in a qualitative study among key populations living with HIV, inclusive of youth, in Tanzania, who articulated the potential to provide relief from daily pill burdens, alleviate stigma, and increase confidentiality [50, 51]. Currently, data on AYLHIV perceptions and feelings on long-acting HIV treatment options in sub-Saharan Africa SA remain limited and are essential to inform future plans for the implementation of this new treatment modality in these unique populations [51].

Our study has several limitations. FGD participants were selected using a convenience sample from Homa Bay among AYLHIV engaged in services at the RCP-supported healthcare facilities, which may result in reporting bias. To remediate this bias, the study was conducted across 60 facilities within 8 Homa Bay sub-counties. The views and perceptions of participating AYLHIV may also not be representative of ALHIV in other settings or contexts. In spite of these limitations, our data summarized unique, detailed experiences, views, and feedback of the AYLHIV in Kenya and guided RCP tools, and service design, and planning for the implementation of future innovations [16, 17, 38, 52]. This project also contributed to the creation of the Committee of African Youth Advisors (CAYA) at EGPAF to assure a mechanism of sustained adolescent and youth engagement when designing and implementing HIV testing and care services for AYLHIV.

## Conclusions

Meaningful engagement of AYLHIV has been a cornerstone in the design and implementation of RCP, from design to tools development, activity implementation, and program evaluation. The qualitative data from our study provides important insight into the needs of the AYLHIV, the barriers and facilitators in accessing HIV testing and care. The AYLHIV feedback described in this study was used in informing, planning, and scaling of the RCP model at a national level in Kenya and in piloting the RCP model in Malawi in 2020 [52–54]. Youth voices informing programmatic work are essential in strengthening our capacity to optimize the support for AYLHIV at all levels, within the community, at schools, and healthcare facilities.

## Supporting information

**S1 Fig. Inclusivity in global research considerations.**
(DOCX)

**S1 Appendix. Research materials.**
(PDF)

## Acknowledgments

We would like to acknowledge and sincerely thank the adolescents and young people who participated in the focus group discussions for their contributions and engagement. Thank you to the dedicated team of research assistants who facilitated the focus group discussions.

Furthermore, we would like to extend our thanks to the staff of Homa Bay County who supported the implementation of the Red Carpet Program.

## Author Contributions

**Conceptualization:** Judith Kose, Tyriesa Howard, Rose Masaba, Job Akuno, Natella Rakhmanina.

**Data curation:** Judith Kose, Tyriesa Howard, Cosima Lenz, Rose Masaba, Job Akuno.

**Formal analysis:** Judith Kose, Tyriesa Howard, Cosima Lenz, Rose Masaba, Job Akuno, Godfrey Woelk.

**Funding acquisition:** Natella Rakhmanina.

**Methodology:** Judith Kose, Tyriesa Howard, Cosima Lenz, Rose Masaba, Job Akuno, Godfrey Woelk, Natella Rakhmanina.

**Project administration:** Judith Kose, Cosima Lenz, Job Akuno, Pieter Leendert Fraaij, Natella Rakhmanina.

**Supervision:** Natella Rakhmanina.

**Validation:** Natella Rakhmanina.

**Writing – original draft:** Judith Kose, Tyriesa Howard, Cosima Lenz.

**Writing – review & editing:** Judith Kose, Tyriesa Howard, Cosima Lenz, Rose Masaba, Job Akuno, Godfrey Woelk, Pieter Leendert Fraaij, Natella Rakhmanina.

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
