## [Editor Report · Decision Letter 0]

26 Dec 2022

PONE-D-22-34176Experiences of Adolescents and Youth with HIV Testing and Linkage to Care through Red Carpet Program in KenyaPLOS ONE

Dear Dr. Kose,

Thank you for submitting your manuscript to PLOS ONE. After careful consideration, we feel that it has merit but does not fully meet PLOS ONE’s publication criteria as it currently stands. Therefore, we invite you to submit a revised version of the manuscript that addresses the points raised during the review process.

We look forward to receiving your revised manuscript.

Kind regards,

Amee Schwitters, PhD

Academic Editor

PLOS ONE

Journal Requirements:

Additional Editor Comments:

Your manuscript covers an important area of research within HIV prevention and treatment, however as currently written the methodology is incomplete to a degree that it makes it difficult to judge the scientific soundness. The methods section needs the following added: detailed description of how participants were recruited (beyond convenience sample) (i.e. sample size determination, male/female/non-binary sample size distribution, etc). Where did focus groups take place, how were transcripts reviewed (one/multiple people), were codes created a priori or after, what was done to minimize bias amongst interviewed/FG leader, etc. As of now, it would be very difficult to replicate your study given the sparsity of details included, on: sampling strategy, data collection procedures, data analysis, and potential sources of bias. Your objectives should also be defined more clearly - what the problem is, why you undertook your research (including why you chose the methodology you used), and what it contributes to our existing knowledge base.

---

## [Author Response · Author response to Decision Letter 0]

21 Feb 2023

Amee Schwitters, PhD

Academic Editor

PLOS ONE 

February 21, 2023

Dear Dr. Amee Schwitters,

Thank you for the opportunity to revise this manuscript: PONE-D-22-34176 “Experiences of Adolescents and Youth with HIV Testing and Linkage to Care through Red Carpet Program in Kenya”. We would like to thank the reviewer for the comments and suggestions and have revised our manuscript and submission accordingly. Please find our point-by-point responses in bold italics to the reviewers’ comments below. 

Reviewers’ Comments:

Please amend your authorship list in your manuscript file to include author P.L. (Pieter) Fraaij.

Thank you for this comment. We have added Pieter Fraaij as an author on the manuscript file. 

We noted that the global inclusivity questionnaire you provided appears to be the unfinished version including struck-through statements and sticky notes throughout the document. Please ensure to provide a finalized, unmarked version of the questionnaire. Thank you for your attention to this request.

We have amended and updated the version of the inclusivity questionnaire and ensure it is unmarked and verified. 

We note your response to our data query as follows:

"We would continue to indicate the data is available upon request as we would like to ensure the continued protection of the privacy and confidentiality of the participants of the study."

And that your current Data Availability statement reads as follows:

"To protect the privacy and confidentiality of participants in the study, the data will not be shared publicly. In response to reasonable requests with the corresponding author de-identified data may be shared in line with ethical approval."

However, PLOS does not allow authors to serve as the sole point of contact for data access requests. Please provide a phone number, email, or other contact information for a non-author or institutional contact which an interested party may contact to acquire the data. We are unable to proceed with an author listed as the sole contact in your Data Availability statement.

Thank you for this comment. We have identified a non-author and institutional contact to be the point of contact pertaining to data access requests. 

Please, let us know if you require any additional information/clarifications. We are grateful for the opportunity to resubmit the manuscript. 

Sincerely, 

Dr. Judith Kose, MD

---

## [Decision Letter · Decision Letter 1]

30 Oct 2023

PONE-D-22-34176R1Experiences of Adolescents and Youth with HIV Testing and Linkage to Care through Red Carpet Program in KenyaPLOS ONE

Dear Dr. Kose,

Thank you for submitting your manuscript to PLOS ONE. After careful consideration, we feel that it has merit but does not fully meet PLOS ONE’s publication criteria as it currently stands. Therefore, we invite you to submit a revised version of the manuscript that addresses the points raised during the review process.

It is clear this is an important research focus, and manuscript. I commend you.

However, the current manuscript requires a lot of corrections as indicated by our Reviewers.

The goal of this manuscript is unclear. Ensure the introduction-results-discussion-conclusion are all in sync. Avoid encumbrances from the larger project.

Provide more details on the preparation of the fieldwork team

Provide more details on the data collection sessions, step by step. including venue characteristics, length, facilitator/s, data capture management and storage during fieldwork and so on

Provide more details on data analysis steps. You need a deeper data analysis e.g. do the codes and sub-codes differ by participants' demographics? If yes, what is your interpretation?

Remember to provide source of quotes appropriately and consider providing then in the narrative only.

Consider using a conceptual framework too.

Add the study tools as Supplementary files

We look forward to receiving your revised manuscript.

Kind regards,

Violet Naanyu, PhD

Academic Editor

PLOS ONE

Journal Requirements:

Additional Editor Comments:

It is clear this is an important research focus, and manuscript.

However, the current manuscript requires a lot of corrections as indicated by our Reviewers.

The goal of this manuscript is unclear. Ensure the introduction-results-discussion-conclusion are all in sync. Avoid encumbrances from the larger project.

Provide more details on the preparation of the fieldwork team

Provide more details on the data collection sessions, step by step. including venue characteristics, length, facilitator/s, data capture management and storage during fieldwork and so on

Provide more details on data analysis steps. You need a deeper data analysis e.g. do the codes and sub-codes differ by participants' demographics? If yes, what is your interpretation?

Remember to provide source of quotes appropriately and consider providing then in the narrative only.

Consider using a conceptual framework too.

Add the study tools as Supplementary files

Reviewers' comments:

Reviewer's Responses to Questions

**Comments to the Author**

1. If the authors have adequately addressed your comments raised in a previous round of review and you feel that this manuscript is now acceptable for publication, you may indicate that here to bypass the “Comments to the Author” section, enter your conflict of interest statement in the “Confidential to Editor” section, and submit your "Accept" recommendation.

Reviewer #1: (No Response)

Reviewer #2: (No Response)

2. Is the manuscript technically sound, and do the data support the conclusions?

Reviewer #1: Partly

Reviewer #2: Yes

3. Has the statistical analysis been performed appropriately and rigorously? 

Reviewer #1: N/A

Reviewer #2: Yes

4. Have the authors made all data underlying the findings in their manuscript fully available?

Reviewer #1: No

Reviewer #2: Yes

5. Is the manuscript presented in an intelligible fashion and written in standard English?

Reviewer #1: Yes

Reviewer #2: Yes

6. Review Comments to the Author

Reviewer #1: Adolescents and Youth living with HIV (AYLHIV) are an important target group in the field of HIV regarding treatment issues. The research gap is presented clearly, and research with, rather than about AYLHIV is commendable.

However, the study is solely descriptive and appears superficial. No underlying theory seems to have been used, nor is there any indication of a deeper analysis of the data. While the FGD guide is not provided, one suspects that central questions in the guide are summarized as themes, thus using a "bucket approach" rather than performing an in-depth analysis of participants' responses. For instance, no analysis seems to have been conducted on any nuanced differences related to participants' gender, age or location which may have brought out some interesting aspects.

Methods Section

- Did the venue of conducting FGDs in clinics affect participants' sharing about service delivery, cf. lines 249.250.

- Can you provide more details about the training? Who trained research assistants, in what areas and for how long?

- Anonymous questionnaires are mentioned - was the socio-demographic data not collected for each respective participant but rather anonymously for the whole group? This seems rather strange. In qualitative research one usually uses pseudonymization rather than anonymization. This would also provide some more background for the quotes (gender, age,etc.).

- Can the FGD guide and the questionnaire be provided?

- The analysis is not explained in sufficient detail. How was the data analyzed? Was it done deductively using the FGD guide questions or inductively? What analytical method was used?

- A more thorough analysis should be done that takes into account the different age groups, gender, location, etc., whether participants were speaking about their own experience or generally about behavior of their gender group or the opposite gender group, their own age group or other groups.

- Was any other source of information used to triangulate findings - interviews, participant observation, document research?

Results Section

-Quotes are not referenced at all - no age or gender of the participant is provided. These should be added.

-I would not use a table for quotes but rather weave them into the text. This would make it a lot more analytical than simply listing quotes under certain sub-themes. Also, some quotes are provided in the text, others in the table.

-Mention clearly whether statements like "certain testing practices were described as problematic, especially among adolescent males" (lines 142.143) are based on male participants' contributions or female participants' contributions - is this an insider or outsider view? Were there differences in the perceptions of male and female participants, of younger and older male participants, etc.?

Discussion

- Lines 327 to 331 could be shifted to the Introduction

- Good to have taken up the issue of "drug-free schools" in the discussion - here it would have been beneficial to have had the view of school principals as well or to critically engage with policies on stopping drug abuse in and around school in Kenya, https://nacada.go.ke/statement-status-alcohol-and-drug-abuse-schools-kenya in the light of adolescents living with chronic infections or diseases.

- Stigma could have been discussed in its various forms, disclosure could have also been discussed in more depth, e.g., the experience of being disclosed to as a perinatally infected youth and self-disclosure to others, the relationship between disclosure and anticipated or enacted stigma. It might improve the article to focus on fewer themes and to discuss them in more depth.

-The discussion mentions implementation steps taken after the study. What is the purpose of the article then? To show the issues adolescents and young adults face when living with HIV or to demonstrate which measures have improved adherence? If the latter is the case, then the whole article could have been written from a perspective of tailored service provision. If the former is the case, a more nuanced article would contribute more significantly to our understanding of adolescents and young adults living with HIV.

Language

- The article should be checked by a native speaker in terms of grammar and word choice.

I would recommend an in-depth analysis of the data and a rewriting of this article.

Reviewer #2: Apart from few typographical errors, the manuscript is good. Consider updating some references, so that your data, for example of HIV prevalence is current.

7. PLOS authors have the option to publish the peer review history of their article (what does this mean?). If published, this will include your full peer review and any attached files.

Reviewer #1: No

Reviewer #2: No

---

## [Author Response · Author response to Decision Letter 1]

15 Dec 2023

Response to reviewers: Manuscript PONE-D-22-34176

Thank you to all reviewers for the opportunity to revise the manuscript PONE-D-22-34176 “Experiences of Adolescents and Youth with HIV Testing and Linkage to Care through Red Carpet Program (RCP) in Kenya”. 

We would like to thank the reviewers for their comments and have revised our manuscript accordingly. 

Please find our point-by-point responses to the reviewers’ comments below. 

Reviewers’ Comments

Reviewer #1

It is clear this is an important research focus, and manuscript. I commend you. However, the current manuscript requires a lot of corrections as indicated by our Reviewers.

The goal of this manuscript is unclear. Ensure the introduction-results-discussion-conclusion are all in sync. Avoid encumbrances from the larger project.

Response: Thank you for this guidance. We have revised the framing and wording of the content of the manuscript to facilitate flow and clarify aligned aims throughout the writeup. 

Provide more details on the preparation of the fieldwork team. 

Response: We have added additional details pertaining to the preparation, training, and sensitization of the field team in the methods section. 

Provide more details on the data collection sessions, step by step. including venue characteristics, length, facilitator/s, data capture management and storage during fieldwork and so on

Response: Thank you for making these valuable points. We have reviewed the methodology and added details such as venue characteristics, length of the sessions, facilitators and data capture management and described in detail the data capture and storage during the fieldwork. 

Provide more details on data analysis steps. You need a deeper data analysis e.g. do the codes and sub-codes differ by participants' demographics? If yes, what is your interpretation?

Response: We have revised the methods and added more details on participant demographics and robust information employed in data analysis undertaken, including on thematic coding. 

Remember to provide source of quotes appropriately and consider providing then in the narrative only.

Response: We have added the source of the quotations as suggested. We have decreased the use of the tables, but also kept table format for many quotes to keep the manuscript narrative within the acceptable word limits and for the purpose of better organizing the quotations by theme. 

Consider using a conceptual framework too.

Response: Thank you for this suggestion. We have added information on our hypothesis and rationale for documenting this work. This study was not designed aligned with a conceptual framework but is rather focused on evaluating the experiences of newly diagnosed AYLHIV linked to care at RCP facilities to assess quality of services and identify areas for optimization. 

Add the study tools as Supplementary files.

Response: Thank you for this comment. We have added the additional research materials to the supplementary materials. 

Reviewer #1: 

Adolescents and Youth living with HIV (AYLHIV) are an important target group in the field of HIV regarding treatment issues. The research gap is presented clearly, and research with, rather than about AYLHIV is commendable. - Thank you. 

However, the study is solely descriptive and appears superficial. No underlying theory seems to have been used, nor is there any indication of a deeper analysis of the data. While the FGD guide is not provided, one suspects that central questions in the guide are summarized as themes, thus using a "bucket approach" rather than performing an in-depth analysis of participants' responses. For instance, no analysis seems to have been conducted on any nuanced differences related to participants' gender, age or location which may have brought out some interesting aspects. 

Response: Our qualitative study had a goal to evaluate youth perspectives through FGDs and, therefore, the sample size was relatively small for the nuanced analysis. We did not aim to analyze the FGDs based on gender, age or location. 

Methods Section

Did the venue of conducting FGDs in clinics affect participants' sharing about service delivery, cf. lines 249.250.

Response: We have added the description of the venues for FGDs. They were similar across the sites. 

Can you provide more details about the training? Who trained research assistants, in what areas and for how long?

Response: We added the details of the training. 

Anonymous questionnaires are mentioned - was the socio-demographic data not collected for each respective participant but rather anonymously for the whole group? This seems rather strange. In qualitative research one usually uses pseudonymization rather than anonymization. This would also provide some more background for the quotes (gender, age, etc.). 

Response: We collected self-identified age ranges for demographics and gender at an aggregate level for FGD participants. This information was added for context to the paper. 

Can the FGD guide and the questionnaire be provided? Response: Yes, we added it as an appendix. 

The analysis is not explained in sufficient detail. How was the data analyzed? Was it done deductively using the FGD guide questions or inductively? What analytical method was used? 

Response: We have added more details clarifying the data analysis. 

A more thorough analysis should be done that takes into account the different age groups, gender, location, etc., whether participants were speaking about their own experience or generally about behavior of their gender group or the opposite gender group, their own age group or other groups.

- Was any other source of information used to triangulate findings - interviews, participant observation, document research?

Response: We have clarified the sources used to triangulate findings. 

We have thoroughly revised and added to the methods section to address the training, preparation, and analysis inquiries posed. 

Results Section

Quotes are not referenced at all - no age or gender of the participant is provided. These should be added.

Response: We have added the self-identified age ranges. We only collected gender information at an aggregated level and added that information. Please, see explanation above. 

I would not use a table for quotes but rather weave them into the text. This would make it a lot more analytical than simply listing quotes under certain sub-themes. Also, some quotes are provided in the text, others in the table.

Response: We have decreased the use of the table but retained some of the tables to preserve the acceptable length of the narrative and for the purpose of better organizing the quotations by theme. 

Mention clearly whether statements like "certain testing practices were described as problematic, especially among adolescent males" (lines 142.143) are based on male participants' contributions or female participants' contributions - is this an insider or outsider view? Were there differences in the perceptions of male and female participants, of younger and older male participants, etc.? 

Response: We collected gender information at an aggregate level, please, see explanation above. 

Discussion

Lines 327 to 331 could be shifted to the Introduction – We shifted as suggested. 

Good to have taken up the issue of "drug-free schools" in the discussion - here it would have been beneficial to have had the view of school principals as well or to critically engage with policies on stopping drug abuse in and around school in Kenya, https://nacada.go.ke/statement-status-alcohol-and-drug-abuse-schools-kenya in the light of adolescents living with chronic infections or diseases. 

Response: We did not engage with school principals in this study but agree their perspective would be beneficial to gage on this topic. 

Stigma could have been discussed in its various forms, disclosure could have also been discussed in more depth, e.g., the experience of being disclosed to as a perinatally infected youth and self-disclosure to others, the relationship between disclosure and anticipated or enacted stigma. It might improve the article to focus on fewer themes and to discuss them in more depth. 

Response: We have added additional descriptions of stigma and its connection to disclosure in the discussion as recommended. 

The discussion mentions implementation steps taken after the study. What is the purpose of the article then? To show the issues adolescents and young adults face when living with HIV or to demonstrate which measures have improved adherence? If the latter is the case, then the whole article could have been written from a perspective of tailored service provision. If the former is the case, a more nuanced article would contribute more significantly to our understanding of adolescents and young adults living with HIV. – 

Response: We conducted this study as we implemented the Red Carpet project. We have clarified that the findings from this study helped us to shape the program to better meet youth needs. We have a separate manuscript in progress analyzing in-depth outcomes of the Red Carpet project. The purpose of this manuscript it to share the Kenyan youth experiences in the four domains of living with HIV and highlight the significance on the meaningful engagement of adolescents in the design of the clinical services. 

Language

The article should be checked by a native speaker in terms of grammar and word choice.

Response: We have reviewed and addressed grammar and word choice with the team of US based native English speakers. 

I would recommend an in-depth analysis of the data and a rewriting of this article.

Response: We have reviewed the analysis and significantly revised the manuscript as a whole. 

Review #2

Apart from few typographical errors, the manuscript is good. Consider updating some references, so that your data, for example of HIV prevalence is current.

Response: Thank you for these comments. We have updated the global prevalence numbers as well as updated outdated references. 

Please, let us know if there is any additional information required. We sincerely hope that this revised version can be considered for publication and thank you for the review which significantly strengthened the manuscript. 

Respectfully, 

Dr. Judith Kose, MD

---

## [Editor Report · Decision Letter 2]

20 Dec 2023

Experiences of adolescents and youth with HIV testing and linkage to care through the Red Carpet Program (RCP) in Kenya

PONE-D-22-34176R2

Dear Dr. Kose,

We’re pleased to inform you that your manuscript has been judged scientifically suitable for publication and will be formally accepted for publication once it meets all outstanding technical requirements.

Kind regards,

Violet Naanyu, PhD

Academic Editor

PLOS ONE
---

## [Editor Report · Acceptance letter]

10 Jan 2024

PONE-D-22-34176R2 

PLOS ONE

Dear Dr. Kose, 

I'm pleased to inform you that your manuscript has been deemed suitable for publication in PLOS ONE. Congratulations! Your manuscript is now being handed over to our production team.

Kind regards, 

on behalf of

Prof. Violet Naanyu 

Academic Editor

PLOS ONE